# Learning to Regrasp by Learning to Place

**Shuo Cheng[1], Kaichun Mo[2], Lin Shao[2]**

[1]University of California, San Diego — scheng@eng.ucsd.edu
[2]Stanford University — {kaichunm, lins2}@stanford.edu

**Abstract:** In this paper, we explore whether a robot can learn to regrasp a diverse set of objects to achieve various desired grasp poses. Regrasping is needed whenever a robot's current grasp pose fails to perform desired manipulation tasks. Endowing robots with such an ability has applications in many domains such as manufacturing or domestic services. Yet, it is a challenging task due to the large diversity of geometry in everyday objects and the high dimensionality of the state and action space. In this paper, we propose a system for robots to take partial point clouds of an object and the supporting environment as inputs and output a sequence of pick-and-place operations to transform an initial object grasp pose to the desired object grasp poses. The key technique includes a neural stable placement predictor and a regrasp graph based solution through leveraging and changing the surrounding environment. We introduce a new and challenging synthetic dataset for learning and evaluating the proposed approach. We demonstrate the effectiveness of our proposed system with both simulator and real-world experiments. More videos and visualization examples are available on our project webpage https://sites.google.com/view/regrasp.

**Keywords:** Regrasping, Object Placement, Robotic Manipulation

## 1  Introduction

*"Regrasping must be performed whenever a robot's grasp of an object is not compatible with the task it must perform."* [1]

Through regrasping, an object can be picked up initially, placed down on an intermediate stable state possibly supported by a second object or the environment geometry, and then be grasped in a target pose and position. Regrasping is a common daily task and plays a crucial role in connecting various manipulation operations to solve complex long-horizon tasks. For example, when assembling parts to an IKEA chair, we may need to pick-and-place these parts several times to find appropriate orientations before mating and putting them together. When preparing the food, we may need to pick-and-place the knife and spatula several times to find the desired grasp poses for tool usages. The ability of a robot to regrasp a wide variety of objects is of tremendous importance for many application domains such as manufacturing or domestic services. However, the large diversity of geometry in everyday objects and the high dimensionality of the state and action space make this a challenging manipulation task. In this paper, we enable a robot to construct a sequence of pick-and-place operations to achieve various desired grasp poses.

Picking objects, or grasping, has attracted great attention in robotics [2, 3, 4, 5, 6, 7]. In contrast, regrasping, which is generating the sequence of pick-and-place operations, has received considerably less attention. Early work [1, 8] builds a Grasp-Placement (GP) to search for regrasp sequences. It first fills valid grasp and placement pairs in the GP table and then searches the table to find a sequence of pick-and-place motion. There are many works following this direction [9, 10, 11]. It usually requires a full observability of the environment and takes a significantly long time before finding a solution given the high-dimensionality of the searching space. Jiang et al. [12] use Support Vector Machines with hand-designed features to score the placement suitability of candidate poses and they focus on the object placement. In this paper, we propose a novel stable object pose prediction model with deep-learned features extracted directly from the raw point clouds and consider both picking and placing operations for regrasping tasks.

Taking as inputs partial observations of the object to regrasp and the surrounding environment, our system predicts a pick-and-place sequence to reach the desired object grasp poses as the output. The

5th Conference on Robot Learning (CoRL 2021), London, UK.

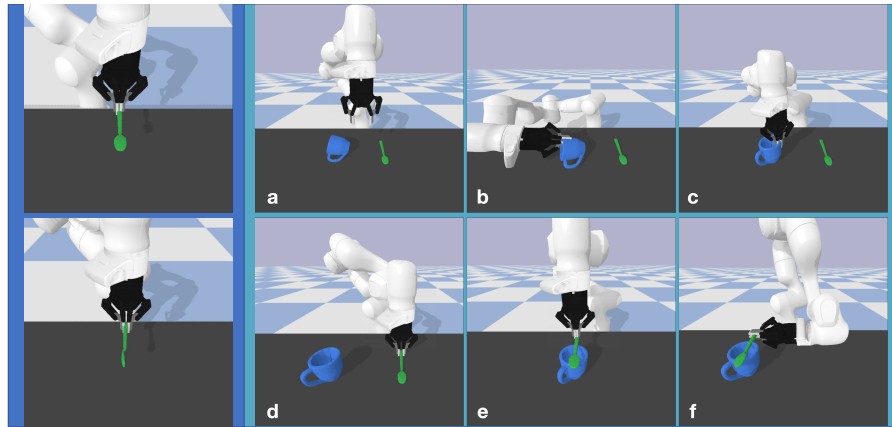

Figure 1: Regrasping must be performed if an initial grasp pose (top-left) is not suitable for downstream task. Taking as inputs the partial observation of the object (the green spoon) and the surrounding environment (the ground and a blue mug), as well as the desired target grasp pose that is more suitable for the downstream task (bottom-left), our system helps robot to adjust the environment and achieve the regrasping task through the following steps: a) Place the spoon because the initial grasp is not reachable to the target grasp pose; b) Grasp the mug, c) Place the mug to construct a stable supporting environment, d) Grasp the spoon, e) Place the spoon with stable pose that is feasible for regrasping, f) Grasp the spoon by up and down for matching the target grasp pose.

system contains a learned object placement module that is able to efficiently generate a diverse set of valid object poses for placement, and a regrasp framework that autonomously creates a regrasp graph to search for desired grasp poses by leveraging and changing the surrounding environment.

Our primary contributions are: 1) proposing a novel manipulation system that enables robots to regrasp a diverse set of objects through actively leveraging and changing the surrounding environment, 2) introducing a learning-based model to predict stable object placement poses; 3) generating an annotated dataset for object regrasping tasks and conducting extensive experiments to demonstrate the effectiveness of our proposed approach.

## 2 Related Work

We review literature related to the two key components in our approach - predicting object placement and planning regrasp graph, and describe how we are different from previous works.

### 2.1 Object Placement

Pick-and-place is one of the most common robotic manipulation tasks. Picking objects, or grasping, has attracted great attention in robotics. For a broader review of the field on data-driven grasp synthesis, we refer to [13, 7]. In contrast, object placement, which is the process of deciding where and how to place an object, has received considerably less attention. Harada et al. [14] presented an object placement planner to find object placement poses by matching planar surface patches on the object with planar surface patches in the environment. Haustein et al. [15] introduced a motion planning algorithm for robots to place a grasped object in a cluttered environment. Jiang et al. [12] employed Support Vector Machines with hand-designed features to score the placement suitability of candidate poses. Unlike these works either requiring full observability or relying on hand-crafted feature engineering, we propose an end-to-end approach with deep-learned features extracted directly from the raw point clouds.

### 2.2 Regrasp for Object Reorientation

Early works [1, 8] build Grasp-Placement (GP) to search for regrasp sequences, by solving the inverse kinematics, checking all collisions to remove invalid grasp-placement pairs, filling them in the GP table, and finally searching within the table to find a sequence of pick-and-place motion. There are many works following this direction. Rohrdanz and Wahl [9] improved the efficiency of regrasp planning using an evaluated breadth-first-search, rated grasp, and placement quantities. Stoeter et al. [10] replaced the GP table with a space of compatible grasp-placement-grasp triplet and searches

this space to find a sequence of pick-and-place motion. Regrasp is also studied in the context of task and motion planning (TAMP). Lozano-Pérez and Kaelbling [16] presented a framework where a symbolic planner plans a sequence of high-level sub-tasks and a constraint satisfaction problem solver plans low-level operations. A regrasp plan is composed of a symbolic pick-and-place sequence and a set of geometrically feasible placements, grasps, and paths, which requires a precise full observability of the entire system.

Wan et al. [11] presented a regrasp planning component for object reorientation. Given the initial and goal pose of an object, the regrasp planning component finds a sequence of robot postures and grasp configurations that reorient the object from the initial pose to the goal pose. Raessa et al. [17] presented a hierarchical motion planner for planning the manipulation motion to re-pose long and heavy objects. They developed a graph-based planning system that combines both regrasp and in-hand manipulation motion considering external support surfaces. Cao et al. [18] increased the reorientation capability of a pick-and-place regrasp by adding a vertical pin on the working surface and using it as the intermediate location for regrasping. The work most related to ours is [19] where they developed a regrasp planning algorithm considering intermediate stable states on fixture such as box baskets. Unlike these planning based approaches, our learning-based approach do not require full observability and learn features directly from raw sensory data. Our approach also allows robots to actively change other objects in the environment to create extra supports. Besides regrasping, object reorientation can also be achieved through in-hand manipulation [20, 21, 22] or dual-arm manipulation [23, 24].

## 3  Problem Definition

In this section, we formulate the object regrasping problem and introduce necessary concepts we use. Note that there may be various other ways to parameterize the same problem.

We consider a single-arm robot equipped with a two-fingered robotic hand. However, our proposed pipeline can be applied to many other settings, such as multi-arm robots or dexterous robotic grippers, to enhance their manipulation performances. We assume that the segmentation of all objects within the surrounding environments is provided and the objects are graspable for the robotic hand. Note that our approach could also be extended to movable but ungraspable objects in the surrounding environment leveraging non-prehensile manipulation processes such as pushing.

**Notations.**  Let $\mathfrak{O} = \{\mathcal{O}_i\}$ denote the set of objects in the scene, including the target object $\mathcal{O}_t$, supporting objects $\mathcal{O}_{i \neq t}$, and the ground $\mathcal{O}_g$. We use $\mathcal{H}$, $\mathcal{R}$, and $\mathcal{C}$ to denote the robotic hand, the robot, and the robot configuration respectively. Each object $\mathcal{O}_i$ and the robotic hand $\mathcal{H}$ has its own local coordinate frame $\mathcal{T}^i$ and $\mathcal{T}^{\mathcal{H}}$. Their pairwise relative transformations are also known. Let $\mathcal{T}_g^i$ and $\mathcal{T}_g^{\mathcal{H}}$ denote the 6D poses of the object $\mathcal{O}_i$ and the robotic hand $\mathcal{H}$ in the world frame $\mathcal{T}^{\mathcal{O}_g}$.

**Grasping Operation.**  A grasp pose $\mathcal{G}_k^i = (\mathbf{p}_k^i, \mathbf{q}_k^i, \mathbf{d}_k^i)$ of an object $\mathcal{O}_i$ is parameterized by two contact points on the object surface $(\mathbf{p}_k^i, \mathbf{q}_k^i)$ and an approaching direction of the robotic hand $\mathbf{d}_k^i$. All three vectors are described in the object local frame $\mathcal{T}^i$. A grasping operation $\mathscr{G}_k^i$ depends on the grasp pose $\mathcal{G}_k^i$, the object pose in the world frame $\mathcal{T}_g^i$, the surrounding environment $\mathfrak{O} - \{\mathcal{O}_i\}$, and the robot's configuration $\mathcal{C}$. A grasping operation $\mathscr{G}_k^i$ is valid if the contact points are in force-closure, reachable by the robot, and occurring no collision in the process. A valid grasp $\mathscr{G}_k^i$ attaches the object $\mathcal{O}_i$ to the robotic hand $\mathcal{H}$, and hence its 6D pose in the world frame can change as the robotic hand moves. We assume all objects, including the supporting objects $\mathcal{O}_{i \neq t}$, can be grasped, since the robot may rearrange the environment in various ways to support the target object $\mathcal{O}_t$ finally.

**Placement Operation.**  When an object $\mathcal{O}_i$ is placed stably on a surrounding environment, we use its global world pose $\mathcal{T}_k^i$ to describe its placement pose. A placement operation $\mathscr{P}_k^i$ of the object $\mathcal{O}_i$ depends on the placement pose $\mathcal{T}_k^i$, the surrounding environment $\mathfrak{O} - \{\mathcal{O}_i\}$, and the robot's configuration $\mathcal{C}$. The placement operation $\mathscr{P}_k^i$ is valid if 1) the placement pose is reachable by the robot, 2) there is no collision, and 3) the object remains stable after the robotic hand releases the object. A valid placement $\mathscr{P}_k^i$ detaches the object $\mathcal{O}_i$ from the robotic hand. Its 6D pose in the world frame $\mathcal{T}_{g,k}^i$ remains when the robot moves away.

**The Regrasping Problem.**  The problem of regrasping object $O_t$ is defined as finding a sequence of valid grasping operations $\mathscr{G}$ and placement operations $\mathscr{P}$ over all objects $\{\mathcal{O}_i\}$ to reach a target grasp pose $\mathcal{G}_*^t$ for the target object $\mathcal{O}_t$.

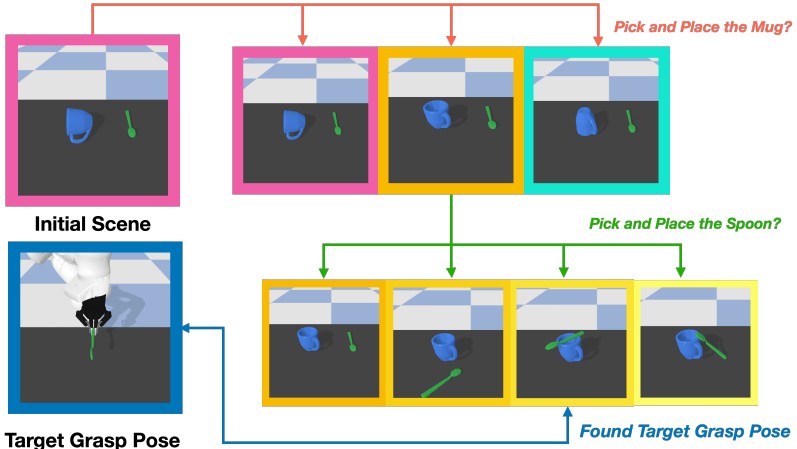

Figure 2: Illustration of our overview pipeline that enables robot to achieve the regrasping task through adjusting the environment step by step. Given the partial observation of all objects and the surrounding environment, our system leverages the learned neural network to propose a diverse distribution of object poses for stable placement, and it utilizes a pose classifier to remove unstable pose proposals. Our system gradually constructs a search graph for finding a valid sequence of object pick and place operations to allow the robot reach the final target grasp pose.

## 4 Technical Approach

In this section, we introduce our regrasping pipeline, consisting of a plan searching algorithm over a regrasp graph (Fig. 2) and a deep-learned model for stable pose placement prediction (Fig. 3).

### 4.1 Regrasp Graph Construction and Searching

A regrasping problem solves a sequence of valid grasping $\mathscr{G}$ and placement $\mathscr{P}$ operations to reach a target grasp pose $\mathcal{G}_*^t$ of the object $\mathcal{O}_t$. We also allow robots to change other objects' poses $\{\mathcal{T}_g^{i \neq t}\}$ by applying several intermediate grasping-and-placing operations, since changing the poses of other objects rearranges the environment and may provide extra supports that can increase the number of valid grasping operations for the target object $\mathcal{O}_t$. Below, we describe in details how to construct a regrasp graph and run a searching algorithm over it to figure out a plan.

**Graph Construction.** Our system creates and leverages a regrasp graph to solve the regrasping problem. Each node $\mathcal{N}$ in the regrasp graph includes stable poses $\{\mathcal{T}_g^i\}$ for all objects $\{\mathcal{O}_i\}$. An edge $\mathcal{E}_{uv}$ connects two nodes $\mathcal{N}_u$ and $\mathcal{N}_v$, if there is one and only one object $\mathcal{O}_j$ whose stable pose is changed between the two nodes, and there exists a valid pair of grasping operation $\mathscr{G}_k^j$ and placement operation $\mathscr{P}_k^j$ that transform the object's stable pose. The edge indicates that the robot can first grasp the object $\mathcal{O}_j$ from its stable pose under one node and then place it at the stable pose in the other node.

Initially, all objects $\{\mathcal{O}_i\}$ are stably placed in the scene. All objects' initial poses define the first node in the regrasp graph. For each object $\mathcal{O}_j$, we use a deep-learned stable object placement pose prediction model (Sec. 4.2) to estimate the object's alternative stable poses under the current surrounding environment. Each alternative stable pose of the object $\mathcal{O}_j$, along with other objects' stable poses, leads to a new node in the regrasp graph. Whenever a new node is created, our system checks whether there are edges connecting it to the rest nodes and add these edges to the graph.

**Graph Searching.** Leveraging the constructed regrasp graph, the solution of a regrasp problem is a path from a starting node $\mathcal{N}_0$ to a target node $\mathcal{N}_t$ that contains a stable object placement pose $\mathcal{T}_g^t$ allowing a valid grasping operation $\mathscr{G}_l^t$ to achieve the target grasp pose $\mathcal{G}_*^t$ for the target object $\mathcal{O}_t$. We use depth-first-search algorithm to search for a valid path.

### 4.2 Stable Object Poses Prediction

We train a neural network to predict stable object placement poses under diverse surrounding environments. Estimating stable placement poses for a wide variety of objects with diverse and compli-

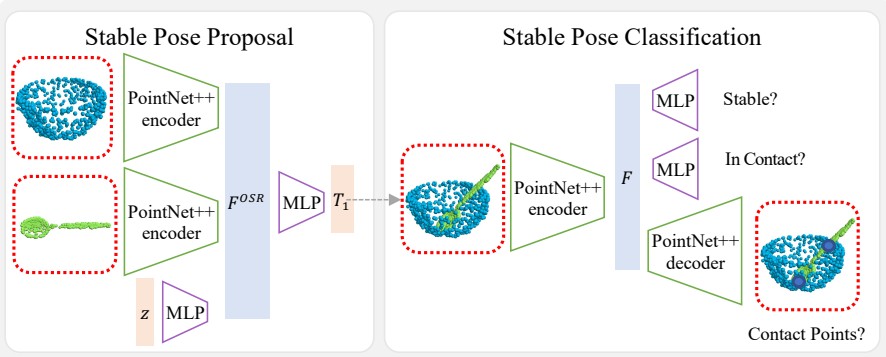

Figure 3: Our proposed framework for predicting stable object placement poses. As shown in the left figure, we first learn a stable pose proposal network that takes as inputs the partial point clouds of the object and the surrounding environment, and outputs a diverse set of rough pose estimations. We formulate it as a generative model and feed in different Gaussian noise vectors $z$'s to query diverse pose prediction outputs. On the right figure, we then show our stable pose classification and refinement network that learns to classify if an object 6D pose leads to a stable placement. We then apply *CEM* [25] to perform local optimization for the pose refinement. We also leverage multi-task training by predicting the contact points between the object and environment as we find it beneficial.

cated geometry is a challenging task. It requires very accurate pose predictions, since a slight error may easily result in an undesired object falling or penetration between the object and its surrounding environment. Hence, we design a two-stage pipeline (Fig. 3) that first predicts diverse rough poses and then refines to obtain more accurate results. We describe the detailed network designs below.

**Stable Poses Proposal.** Given an object $\mathcal{O}_i$ and a static surrounding environment $\mathcal{S} = \mathfrak{O} - \{\mathcal{O}_i\}$, we first train a generative model that learns to propose a diverse set of poses $\{\mathcal{T}_k^i\}$ for the object $\mathcal{O}_i$ to be stably placed in the environment $\mathcal{S}$.

Fig. 3 (left) illustrates the inputs, outputs, and design overview for this network. In our implementation, we represent both the partial observations of the object $X_i \in \mathbb{R}^{m \times 3}$ and surrounding environment $X_\mathcal{S} \in \mathbb{R}^{m \times 3}$, captured by the robot's RGB-D camera, as 3D point clouds consisting of $m = 1,024$ points. We first employ two PointNet++ encoders [26] to extract global point cloud features $f_i \in \mathbb{R}^{128}$ and $f_\mathcal{S} \in \mathbb{R}^{128}$. To propose multiple solutions to the stable poses of the object $\mathcal{O}_i$, we sample a random Gaussian noise $z \in \mathbb{R}^3$ as an additional input and use an Multilayer perceptron (MLP) to extract a feature $f_z \in \mathcal{R}^{64}$. We concatenate $f_i$, $f_\mathcal{S}$, and $f_z$ to produce a single joint feature $f_{i,\mathcal{S},z}$ and finally employ another MLP to decode a final 6-DoF pose $\hat{\mathcal{T}}_{g,z,\mathcal{S}}^i$ for the object $\mathcal{O}_i$. For brevity, we omit the subscripts $g$ denoting the global world coordinate, $\mathcal{S}$ indicating the environment, and the superscript $i$ representing the shape $\mathcal{O}_i$ in the following paragraph. The 6-DoF pose is composed of a 3D translation and a 3D orientation. For the orientation part, we adopt the axis-angle representation, which has been shown to be effective for pose prediction task [27].

Sampling different Gaussian noises $\{z_l\}_{l=1}^r$, we obtain a diverse set of pose predictions $\{\hat{\mathcal{T}}_{z_l}\}_{l=1}^r$. Provided with a ground-truth list of stable poses $\{\mathcal{T}_{z_l}\}_{l=1}^{r_*}$ generated by running several random interaction trials in physical simulation, we define a loss $\mathcal{L}_M$ to train our pose prediction outputs.

$$\mathcal{L}_M(\{\hat{\mathcal{T}}_{z_l}\}_{l=1}^r, \{\mathcal{T}_{z_{l'}}\}_{l'=1}^{r_*}) = \sum_{l=1}^r \min_{l'} L(\hat{\mathcal{T}}_{z_l}, \mathcal{T}_{z_{l'}}) + \sum_{l'=1}^{r_*} \min_l L(\hat{\mathcal{T}}_{z_l}, \mathcal{T}_{z_{l'}}) \qquad (1)$$

where $L$ measures the distance between two 6-DoF poses. We implement $L$ as the average $L_2$ distance between the corresponding points of the object point clouds after applying the two poses.

**Stable Pose Classification and Refinement.** We train a model to classify the stability of the sampled poses. For each predicted pose $\hat{\mathcal{T}}_{z_l}$, we first concatenate the transformed object point cloud $\hat{\mathcal{T}}_{z_l}(X_i)$ with the surrounding environment point cloud $X_\mathcal{S}$. Then, we augment this combined point cloud with one extra 1D arrays of ones and zeros, to indicate if a point belongs to $X_i$ or $X_\mathcal{S}$, along the XYZ dimension to form a tensor of shape $(M + M, 3 + 1)$. With this augmented input, a straightforward method for stable classification would be using a PointNet++ [26]. However, we find that

this naive approach does not provide satisfying results due to the intrinsic difficulty of analyzing stability among two separate geometry components.

To facilitate learning, we propose to solve this problem via a multi-task supervised learning approach. After obtaining the feature from the PointNet++ backbone, we have three task branches that perform stable classification, contact classification, and contact point regression simultaneously. For contact point regression, we use a PointNet++ decoder to regress the 3D offset $v_i$ from each point $p_i$ of the input point cloud to its nearest ground-truth contact point with a loss defined as:

$$\mathcal{L}_{offset} = \sum_{p_i \in X} L(p_i + v_i, \arg\min_{c \in C} \|c - p_i\|_2) \tag{2}$$

Here $X$ denotes the input point clouds including the transformed object point cloud $\hat{\mathcal{T}}_{z_l}(X_i)$ and the environment point cloud $X_\mathcal{S}$. Based on the predicted $\hat{\mathcal{T}}_{z_l}$ pose, we calculate the nearest ground truth stable pose from Eqn. 1 and collect the corresponding contact point sets between the object and the environment in the simulation denoted as $C$. We implement $L$ as smooth $L1$ loss. We also adopt a variance loss $L_{variance}$ similar to [27] to reduce the variance of the group $\{p_i + v_i\}$ that correspond to the same contact point. We train two MLP classifiers for stable pose and contact classification with losses denoted as $\mathcal{L}_{stable}$ and $\mathcal{L}_{contact}$ respectively. Our total loss is defined as:

$$\mathcal{L}_C = \mathcal{L}_{stable} + \lambda_1 \mathcal{L}_{offset} + \lambda_2 \mathcal{L}_{contact} + \lambda_3 \mathcal{L}_{variance} \tag{3}$$

where $\mathcal{L}_{stable}$ and $\mathcal{L}_{contact}$ are standard binary cross entropy losses.

Leveraging the above stable pose classifier function denoted as $\mathcal{V}$, we apply *CEM* [25] to search for a locally optimal goal pose starting from the initial predicted poses. We first apply the predicted transformation $\hat{\mathcal{T}}$ to the object point cloud $X_i$ to get a point cloud $\tilde{X}_i$. A point $a$ in the *CEM* searching action space is a 6D transformation $\mathcal{T}_a$ which transforms the object point cloud $\tilde{X}_i$ into $\mathcal{T}_a(\tilde{X}_i)$. A point $s$ in the CEM state space is the transformed object point cloud along with the surrounding environment point cloud $(\mathcal{T}_a(\tilde{X}_i), X_\mathcal{S})$. When selecting the action $a$, we run a derivative-free optimization method CEM [25] to search within the 6D pose space to find a 6D transformation $\mathcal{T}_a$ associated with the highest score in the value model $\mathcal{V}(s)$.

$$a^* = \arg\max_a \mathcal{V}(\mathcal{T}_a(\tilde{X}_i), X_\mathcal{S}) \tag{4}$$

The 6D pose associated with $a^*$ is the final stable placement pose produced by our model.

### 4.3 Object Grasping

In this subsection, we describe how our system performs valid grasping operations. Recall that a grasp pose $\mathcal{G}_k^i$ of the object $\mathcal{O}_i$ is parameterized by two contact points on the object surface $(\mathbf{p}_k^i, \mathbf{q}_k^i)$ and an approaching direction of the robotic hand $\mathbf{d}_k^i$. We adopt UniGrasp [6], an efficient data-driven grasp synthesis method, to generate a list of valid grasp poses $\{\mathcal{G}_k^i\}$ of the object $\mathcal{O}_i$. UniGrasp learns a deep neural network to select a set of contact points $(\mathbf{p}_k^i, \mathbf{q}_k^i)$ from the input point cloud of the object $\mathcal{O}_i$. We refer to Shao et al. [6] for more detailed description of the method. Given two contact points $\mathbf{p}_k^i$ and $\mathbf{q}_k^i$, we calculate a plane perpendicular to the line segment of the two contact points intersecting at the middle point of the line segment. We then consider 36 approaching directions evenly distributed within the plane, which produces 36 potential grasp poses. For each grasp pose $\mathcal{G}_k^i$, we solve the inverse kinematics of the robot. If there is no valid inverse kinematic solution or there are collisions in all possible approaching directions, we drop the contact point pair and continue checking for the next pair of contact points. We keep running this process until we find a valid grasping operation if there exists one.

## 5 Experiments

In this work, we develop a learning framework for robots to learn to regrasp objects. Our experiments focus on evaluating the following questions: (1) How effective is our proposed learning-to-place approach compared to other baselines? (2) Whether our proposed learning-to-regrasp pipeline can find good solutions to the object regrasping problem? (3) Whether the multi-task learning for contact point predictions we use in Sec. 4.2 is useful or not? (4) Whether our proposed pipeline is robust to various sources of noises (e.g. object segmentation, sensing, dynamics)? Please refer to supplementary materials for experiments associated with (3) and (4), ablation study, and failure cases analysis.

**Dataset.** We construct a dataset containing 50 objects (i.e., spoon, fork, hammer, wrench, etc.) and 30 supporting items (i.e., mug, box, bowl). We then split these objects and supporting items into training and test sets. No test objects and supporting items have been seen during training. We generate 249 pairs from the training set of objects and supporting items, and 38 pairs from the test. We collect the placement data for supporting items by randomly placing the supporting items with respect to the initial environment (i.e., the ground), and running the simulation to check whether the supporting items keep static. To make the stable placement robust to variant dynamics and geometry, we randomize the dynamic parameters including friction, mass, and external forces. We describe the details in the supplementary. In total, we generate around one million poses for training set and 15 thousand poses for testing set. We visualize the data in supplementary material.

**Settings.** We first set up the simulation environment in PyBullet [28] for evaluating our full pipeline on synthetic data. We load the object at the predicted pose and check if there is collision and whether the object is stable. We then evaluate the regrasping task on a real robot system.

## 5.1 Object Stable Placement

We evaluate our proposed framework on learning stable placement poses in this subsection.

| method | bowl-fork | bowl-spoon | box-hammer | box-wrench | box-spatula | mug-fork | mug-spoon | mug-corkscrew | mean |
|---|---|---|---|---|---|---|---|---|---|
| *Random* | 0.000 | 0.001 | 0.109 | 0.039 | 0.000 | 0.000 | 0.000 | 0.012 | 0.020 |
| *Jiang et al. [12]* | 0.037 | 0.080 | 0.201 | 0.113 | 0.047 | 0.018 | 0.125 | 0.092 | 0.089 |
| *Jiang et al. [12] + heuristic* | 0.175 | 0.041 | 0.437 | 0.852 | 0.603 | 0.013 | 0.128 | 0.000 | 0.281 |
| *Ours* | **0.869** | **0.848** | **0.894** | **0.911** | **0.905** | **0.674** | **0.834** | **0.829** | **0.845** |

Table 1: Overall comparison of the stable placement accuracy. We report the accuracy of our method and different baseline methods for each category and the average accuracy over all the categories. Our method significantly outperforms these baseline methods.

| method | bowl-fork | bowl-spoon | box-hammer | box-wrench | box-spatula | mug-fork | mug-spoon | mug-corkscrew | mean |
|---|---|---|---|---|---|---|---|---|---|
| *Jiang et al. [12] + heuristic* | 2.500 | 1.000 | 28.000 | 28.333 | 10.333 | 2.000 | 4.250 | 0.000 | 9.552 |
| *Ours* | **16.900** | **18.100** | **45.850** | **49.100** | **52.133** | **14.500** | **15.750** | **12.450** | **28.098** |

Table 2: Overall comparison of the predicted stable placement poses' diversity.

**Evaluation of Object Placement Accuracy.** Our pipeline samples and refines 128 poses for each object and supporting item pair. We use the pose classifier to estimate the scores for these 128 poses and use a threshold of 0.8 to filter out the poses with low scores. To verify the remaining poses, we run a forward simulation in PyBullet [28] to check whether the objects at these poses are stable. The results of the stable placement accuracy for each category are reported in Table 1.

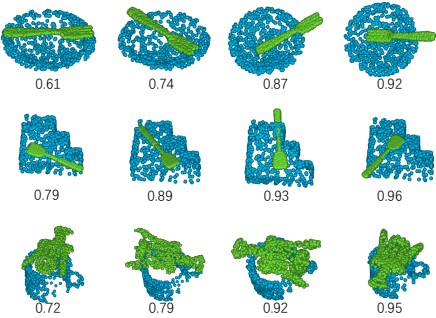

Figure 4: Point cloud visualization of sampled stable poses from our proposal network. Our proposal network can generate diverse stable placements of the candidate object (colored in green). We also mark the predicted stable scores.

We first adopt a baseline (*Random*) which is based on a random pose sampling strategy. The rotation value is sampled from SO(3) uniformly, and translation value is uniformly sampled within this region of [$\pm$0.05, $\pm$0.05, $z$+0.02] in the supporting item's local frame, where $z$ is determined by the maximal height of each supporting item. We use the meter as the unit of translation value throughout the experiment section.

We slightly modify the approach in [12] to work in our setting and use it as another baseline (*Jiang et al. [12]*), in which a stable pose classifier is learned based on hand crafted features. We randomly sample 128 poses in the local frame of the supporting environment, and classify these poses using the stable pose classifier. We evaluate the accuracy of these stable poses in PyBullet [28] and report the results in Table 1. The results suggest that even with random sampled poses, a simple classifier with hand-craft feature can improve the stable placement accuracy. We also design another baseline that adopts heuristic rules for pose sampling but with the same classifier from [12]. For this baseline (*Jiang et al. [12] + heuristic*), the object translation in the horizontal dimensions (XY dimensions)

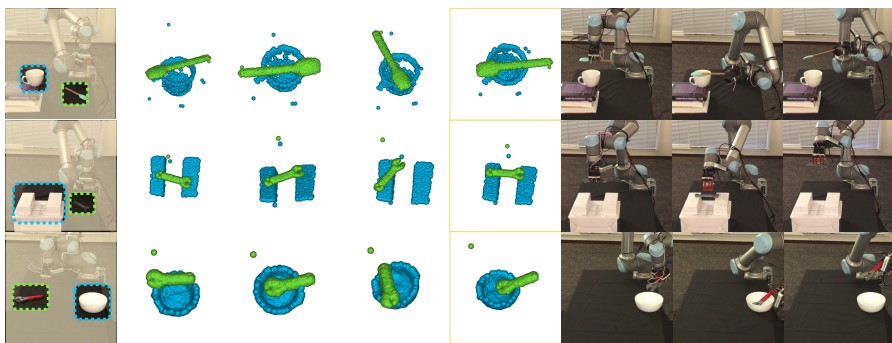

Figure 5: Demonstration of regrasping tasks on real world data and visualization of stable poses generated by our model.

is sampled in the radius of $0.9 * r$ and the translation in the gravity dimension (Z dimension) is determined by $z + 0.01$, where $r$ and $z$ are the radius and maximal height of supporting item respectively. We additionally rotate the object to make the longest axis of the object horizontally. While this baseline can achieve some reasonable results in easy situations when the supporting item has a nearly flat horizontal surface (i.e., box-wrench, box-spatula), it perform worse with supporting items with complicated geometry (i.e., bowl, mug). The results demonstrate that a good pose proposal algorithm is also of great importance to the success of stable object placement. With carefully designed stable pose proposal network and stable pose classifier network, our framework achieves the best results over all the approaches.

**Evaluation of Object Placement Diversity.** We evaluate whether the pose proposal network can generate a reasonable distribution of the stable poses for placing the object over the supporting items. A diversifying stable pose distribution is crucial for the success of finding desirable regrasping solutions. For each input object and supporting item pair, we draw 128 poses from the proposal network, and transform the candidate object to the proposed stable poses. The qualitative results are presented in Fig. 4. Our visualization demonstrates that our proposed stable poses yield diverse object placements.

To quantitatively measure the diversity of the stable object placements, we define two poses as two different poses if their translation difference is larger than $\tau_1$ and the rotation angle between them is larger than $\tau_2$. In our experiment, $\tau_1$ and $\tau_2$ are set to be $0.03m$ and $30°$ respectively. We report the average number of different stable object placements in Table 2. The results showing that although random uniform sampling combined with heuristic rules can generate diverse stable poses for some easier environment (i.e., boxes with flattened surface), it can not handle complicated geometry. Our proposed method outperforms the baseline by a large margin.

### 5.2 Object Regrasping Performance

We evaluate the performance of our model in real-world experiments on everyday objects. We first test the pose proposal network on point clouds from Intel RealSense RGB-D camera. Utilizing the camera matrix, we can reconstruct a 3D point cloud. The point clouds are fed into our model to predict stable placement poses. We then run Unigrasp [6] to generate a list of valid grasp poses and search for the desired target poses based on the pipeline described in Sec. 4. The generated stable placements and several key frames of the regrasping process are visualized in Fig 5. For more results on the regrasping experiments, please refer to our supplementary material.

## 6 Conclusion

We present a system that can regrasp a diverse set of everyday objects to desired grasp poses for various manipulation tasks. Our system learns to predict the stable placement of objects based on partial point clouds of the object and the surrounding environments. The system also creates a regrasp graph and allows robots to leverage and actively change other objects' poses to provide extra supports. We demonstrate the effectiveness of our system on both challenging synthetic dataset and in real world regrasping tasks. Currently our system only takes raw vision sensory data as inputs. For future work, we would like to explore how to incorporate other modalities such as force or tactile signals in our system.

**Acknowledgments**

We would like to thank the UnitX company (https://www.unitxlabs.com) for providing us the UR5 robot arm used in the real robot experiments. We thank Shenli Yuan for helping us set up the customized two-fingered gripper, and Hongzhuo Liang for helping us set up the network configuration.

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
