# OpenReview forum: "Learning to Regrasp by Learning to Place"
_robot-learning.org/CoRL/2021/Conference — CoRL2021 Poster_

### Official Review · Reviewer_FiyG · 2021-07-13

**Originality:** Good
**Technical Quality:** Good
**Clarity Of Presentation:** Very Good
**Impact:** 3

**Recommendation:**

Weak Accept: I recommend accepting the paper, but will not argue for my recommendation if the majority of other reviewers have a different opinion.

**Summary:**

The paper proposes a method to plan sequences of regrasps by learning a pose stability prediction network from partial point cloud observations. Separately, a stable pose proposal network is also learned. Together with an off-the-shelf grasp sampler (Unigrasp), a graph of valid regrasp (grasp and placement poses) can then be constructed. Searching in this graph can generate a sequence of regrasp actions that move an object to a desired target pose. Simulation experiments show a regrasp success rate of 73.3% across 9 object categories, while a baseline using randomly proposed poses achieves 0%.

**Issues:**

Please see the Strengths and Weaknesses section. The website was also empty at the time of review.

**Reviewer Expertise:**

Very good: Comprehensive knowledge of the area

**Strengths And Weaknesses:**

Strengths
-	Using partial pointclouds as input is an improvement over prior works that require full object information.
-	Use of multi-task learning to improve pose stability classification is interesting.
-	The paper is clearly written and easily understood.

Weaknesses
-	I have two main criticisms. One is there does not seem to be a notion of robustness in the stable pose prediction dataset. Two is it was not clear what kind of generalization was being tested aside from the test object pairs.
-	For robustness – some of the visualized stable poses appear quite precarious, with a spoon being placed near the edge of the top of the bowl. It seems quite possible that with small noises in sensing and actuation that some of the predicted stable poses would yield unstable behaviors when executed by a robot. Stable pose data could benefit from perturbations in poses during data generation. Additional noises in object dynamics (friction, mass, and center of mass) and sensing (it was not mentioned whether the partial point clouds were rendered with realistic noise profiles) during data generation would make the method more compelling.
-	For generalization – although pairs of objects were tested, it was not clear whether the pair contained objects that were used in training. For example, if bowl-fork is a test pair, does that mean both bowl and fork were not in the training set, or does that mean the bowl-fork pairing was not in the training set but the objects themselves appeared in other pairings in the training set?
-	The latter case would make the paper more compelling, and additional experiments on generalizing to completely unseen objects, not just object pairs would be appreciated.
-	While the paper mentions using partial pointclouds, in Figure 3 the pointcloud appears to show part of the spoon is visible even though it should be occluded by the front side of the bowl. Is this just for visualization? In reality, if the camera is placed at that angle, then the lower half of the spoon would be occluded by the bowl, so that part of the pointcloud should not be visible.
-	This work also seems to assume that either the objects are completely separated in the beginning so their pointclouds can be obtained separately, or the method has access to a robust pointcloud part segmentation algorithm that can do the same when the objects are close together or in contact. However, there is no discussion of such assumptions in the paper.
-	The video does not show a complete sequence of a robot executing the planned regrasp motions. Having such a video demonstration would more clearly illustrate the proposed method, even if it’s only in simulation.
-	The experiment results are a little hard to interpret. The authors mainly report pose stability prediction accuracies, but not the success rate of the overall regrasping framework. Only an average success rate of 73.3% is given, but more detailed results and analysis here would strengthen the paper.
-	There is also no in-depth discussion of failure modes – are there any patterns to the cases when stability prediction fails? Why does the model perform poorly in some object pairings but not others?


**Summary Of Recommendation:**

My recommendation of weak reject is based on the lack of more in-depth discussions of the problem and presentation of regrasp results.

**Update** I have updated my recommendation to weak accept after the authors' additional experiments and clarifications.

---

### Official Review · Reviewer_bwfS · 2021-07-23

**Originality:** Good
**Technical Quality:** Fair
**Clarity Of Presentation:** Very Good
**Impact:** 4

**Recommendation:**

Weak Accept: I recommend accepting the paper, but will not argue for my recommendation if the majority of other reviewers have a different opinion.

**Summary:**

The paper addresses the problem of placing a target object in a desired pose using a robotic arm, potentially requiring a sequence of grasp and place (regrasp) manipulation of the target object or other supporting objects in the scene. The proposed method employs a leaned module that predicts stable placements of an object in its surrounding, both expressed as point clouds. The placement module generates possible poses which are evaluated by stable, contact, and offset from contact points, and the CEM method is used for selection. A sequence of pick and place is planned by constructing  a regrasp graph by applying grasp and place motions, connecting different nodes corresponding to the stable poses of all the objects in the scene that are accessible through a pick and place of exactly one object, and searching for a path from the initial node to a target node using depth first search. The  method is validated in simulation.

**Issues:**

All issues are described above

**Reviewer Expertise:**

Very good: Comprehensive knowledge of the area

**Strengths And Weaknesses:**

I find the method quite interesting and creative. The problem is very challenging and well motivated. The reported results for both stable placement and overall planned sequence in simulation are also appealing.
However, the proposed method makes some assumptions and is applied in simulation without really reasoning about how this could work in real settings and not just simulation.
The assumption that a perfect segmentation of the scene is provided at each point from which it can be derived which are the objects and which is the target is often not very realistic. I would try to suggest to at least show the robustness of the method to some level of uncertainty.
The data used for stable placement with the accompanied ground truth annotations would be difficult to replicate in real setting. If manual annotations are required or reliance on sim2real capabilities, this should be somehow considered in the paper.
In addition, the regrasp construction and search, which is executed in real-time, could potentially be very computationally demanding. Without any guiding heuristic, the graph can grow very large until the actual target is achieved, and there are no convergence guarantees, at least based on how this process is described in the paper. I think some notion of computation times (construction of nodes and edges) and reasoning about bounds on the grasp size are due. In addition, I would consider incorporating leaning also in this stage as well.

**Summary Of Recommendation:**

The paper describes an interesting approach for solving a challenging and important problem. I think it would be much better after providing some indication about how applicable it would be in real settings

---

### Official Review · Reviewer_nUvc · 2021-07-23

**Originality:** Very Good
**Technical Quality:** Very Good
**Clarity Of Presentation:** Excellent
**Impact:** 3

**Recommendation:**

Strong Accept: I recommend accepting the paper and will argue for my recommendation even if other reviewers hold a different opinion.

**Summary:**

In this paper, the robot's goal is to achieve a desired grasp on an object by constructing a sequence of pick-and-place operations, i.e. a form of regrasping. The paper proposes a novel stable object pose prediction model with deep-learned features extracted directly from raw point clouds. This is used to generate a diverse set of placement poses for a regrasping graph, which is then searched over to find an action sequence. The regrasping graph is constructed of nodes that correspond to stable poses for all of the objects in the scene and edges that correspond to a pick-and-place operation that moves one object. Thus the pose prediction method proposed by the paper is used to generate the nodes, the edges are generated via motion planning, existing work (UniGrasp) is used to generate the grasps, and depth first search is used to find a path through the graph.

The stable object pose prediction framework is a two-part process that first predicts a diverse set of "rough" poses that are then refined via the second stage. The first stage passes partial point clouds into a PointNet++ encoder, along with a Gaussian noise vector to encourage the diverse solutions. A multilayer perceptron outputs a 6-DoF object pose that is used to transform the object's point cloud for the second stage. Rather than directly predicting stability, the paper frames this question as a multi-task supervised learning problem wherein the network branches to predict stability classification, contact classification and contact point regression. The method leverages this classifier and CEM to further refine the 6-DoF stable pose of the object. For training data the paper presents a dataset of pairwise stable placements evaluated in PyBullet. Within the main paper, the framework is evaluated against a baseline strategy that randomly samples placements. With respect to the validation of the stable object pose prediction framework, the experiments first evaluate its ability to generate diverse poses (the first stage) and then evaluate whether the generated poses are stable using PyBullet (the second stage). The entire system is then evaluated, with some ablation studies, across 30 scenes in simulation.

The appendix contains another ablation study, discussions of the collection of real-world point clouds and comparisons to two other baselines from existing work.

**Issues:**

My primary suggestion would be to include the baseline comparisons from the appendix (i.e. the two variants of Jiang et al) into the main paper. These comparisons greatly strengthen the experimental section and many readers could miss them if they are exclusively in the appendix.

A few more minor questions/comments:
- The entire regrasping system is described as having a 73.3% success rate. When the system fails, where are these failures normally coming from? Are poor placements the primary source of failure or is there another aspect?
- The regrasping shown in the video attachment is primarily performed by a human. A more compelling demonstration of the system would include simulation videos of the robot regrasping from the experiments described in Sec 5.2.
- In reviewing the literature on regrasping, it would be worth mentioning other formulations such as in-hand and two-hand regrasping [1, 2, 3, 4].
- With respect to future iterations of the system, it could improve efficiency to incorporate lazy search [5, 6] in the construction and search of the regrasp graph. I would not expect a revised version of this paper to include this modification.
- The regrasping problem in this paper is framed such that there is one target object and any other objects in the scene can be moved in order to accommodate the target object. How could the method be adapted to handle multiple target objects?


[1] Cruciani, Silvia, et al. "Dual-arm in-hand manipulation and regrasping using dexterous manipulation graphs." arXiv, 2019.

[2] Chavan-Dafie, Nikhil, and Alberto Rodriguez. "Regrasping by fixtureless fixturing." IEEE CASE, 2018.

[3] Xue, Zhixing, J. Marius Zoellner, and Ruediger Dillmann. "Planning regrasp operations for a multifingered robotic hand." IEEE CASE, 2008.

[4] Saut, Jean-Philippe, et al. "Planning pick-and-place tasks with two-hand regrasping." IEEE IROS, 2010.

[5] Dellin, Christopher, and Siddhartha Srinivasa. "A unifying formalism for shortest path problems with expensive edge evaluations via lazy best-first search over paths with edge selectors." ICAPS, 2016.

[6] Bohlin, Robert, and Lydia E. Kavraki. "Path planning using lazy PRM." IEEE ICRA, 2000.

**Reviewer Expertise:**

Good: General knowledge of the area

**Strengths And Weaknesses:**

The paper's primary contribution is a novel stable object pose prediction method that operates directly on point clouds. This method is used within a pick-and-place regrasping framework. The paper is clear in stating that the pose prediction method is the main focus and is fairly clear that the regrasping graph structure draws heavily from prior work (which the paper does a good job reviewing over). The problem statement (Section 3) very precisely defines the terminology and problem being tackled. The generality of a pose prediction method that operates directly on point clouds means that this could be broadly applied in other frameworks and applications.

The random sampling strategy baseline that the method is compared against is a very weak baseline, as evidenced in Table 1 by the fact that it often scores 0% (or close to 0%) accuracy. With such low performance, which is to be expected from such a method, it's unclear if it actually serves as a useful comparison. The appendix provides much more reasonable baselines to compare against.

A main strength of the method, as discussed above, is that it operates directly on point clouds, which somewhat frees the method from being as object-centric. The creation of the dataset, as described in Sec 5, seems to suggest that the method only considers placements where the object is supported by one object. Would augmenting the dataset allow for placements where the object is supported by multiple objects?

**Summary Of Recommendation:**

I am weakly recommending the acceptance of this paper. The proposed method to generate stable object poses from point cloud data is sound and has wide applicability. The regrasping aspect of the paper is not overly novel and instead serves as a domain where the pose prediction can be compellingly demonstrated. As mentioned above, the experimental section is weakened by the poor baseline but this can be easily overcome by highlighting the comparisons made in the appendix.

---

### Meta-Review · Area_Chair_Esee · 2021-08-13

**Recommendation:** Accept (Poster)
**Confidence:** 5

**Metareview:**

Reviewers are mixed. Whilst all reviewers are encouraged by the novelty over prior works, two reviewers raise various issues regarding how applicable this method would be in real-world settings, and I would like to hear the authors' opinions on these comments. Whilst real-world experiments are not always essential to demonstrate the importance of a method, some analysis on the robustness to real-world noise, such as segmentation masks, is important to at least show potential for this to work in the real world. Reviewer bwfS also notes how the annotations rely on ground-truth data, which would be difficult to obtain in a real-world setting, and yet sim-to-real experiments are not provided. So, as well as answering queries raised by the reviewers, I would like to see the authors' discuss how applicable the method is to real-world settings, and how they can prove this applicability.

-----

Following the reviews, the authors have now provided evidence of applicability to the real world, with additional real-world experiments and analysis of sensitivity to real-world noise. Authors have also addressed many of the issues raised by the reviewers, and all three reviewers now recommend acceptance of the paper.

---

### Decision · Program_Chairs · 2021-09-13

**Decision:**

Accept (Poster)

**Comment:**

Reviewers are mixed. Whilst all reviewers are encouraged by the novelty over prior works, two reviewers raise various issues regarding how applicable this method would be in real-world settings, and I would like to hear the authors' opinions on these comments. Whilst real-world experiments are not always essential to demonstrate the importance of a method, some analysis on the robustness to real-world noise, such as segmentation masks, is important to at least show potential for this to work in the real world. Reviewer bwfS also notes how the annotations rely on ground-truth data, which would be difficult to obtain in a real-world setting, and yet sim-to-real experiments are not provided. So, as well as answering queries raised by the reviewers, I would like to see the authors' discuss how applicable the method is to real-world settings, and how they can prove this applicability.

-----

Following the reviews, the authors have now provided evidence of applicability to the real world, with additional real-world experiments and analysis of sensitivity to real-world noise. Authors have also addressed many of the issues raised by the reviewers, and all three reviewers now recommend acceptance of the paper.